Ground beetle assemblages inhabiting various age classes of Norway spruce stands in north-eastern Poland

Nietupski Mariusz
Kosewska Agnieszka
Ludwiczak Emilia emilia.ludwiczak@uwm.edu.pl
Department of Entomology, Phytopathology and Molecular Diagnostics, University of Warmia and Mazury in Olsztyn , Olsztyn , Poland
Gillespie Joseph
Electronic publication date: 2023 Dec 1
Publication date: 2023
Volume: 11
Electronic Location ID: e16502
Received 2023 Apr 28; Accepted 2023 Oct 31
Copyright: ©2023 Nietupski et al.
Copyright year: 2023
Copyright holder: Nietupski et al.
License: This is an open access article distributed under the terms of the Creative Commons Attribution License, which permits unrestricted use, distribution, reproduction and adaptation in any medium and for any purpose provided that it is properly attributed. For attribution, the original author(s), title, publication source (PeerJ) and either DOI or URL of the article must be cited.
License URL: https://creativecommons.org/licenses/by/4.0/

Keywords: Age of spruce stands, Carabid beetles, Species diversity, MIB

Funding: The University of Warmia and Mazury in Olsztyn, Faculty of Agriculture and Forestry, Department of Entomology, Phytopathology and Molecular Diagnostics 30.610.010-110 The Ministry of Science and Higher Education within the “Regional Initiative of Excellence” program No. 010/RID/2018/19 The results presented in this article were obtained as part of a comprehensive study financed by the University of Warmia and Mazury in Olsztyn, Faculty of Agriculture and Forestry, Department of Entomology, Phytopathology and Molecular Diagnostics (30.610.010-110). The project was financially supported by the Ministry of Science and Higher Education within the “Regional Initiative of Excellence” program for the years 2019–2023, Project No. 010/RID/2018/19, amount of funding 12.000.000 PLN. The funders had no role in study design, data collection and analysis, decision to publish, or preparation of the manuscript.

==============================
Assemblages of epigeic ground beetles living in Norway spruce forests in north-eastern Poland in three age ranges: young: 20–30 years (A); middle-aged: 40–50 years (B); old: 70–80 years (C) were investigated. In each age category, 4 plots with 5 Barber traps were set up. Ground beetle assemblages were compared in terms of their abundance, species richness, and the Shannon H’ index value. Quantitative ecological description of the carabids captured in the analysed age-classes of Norway spruce forests was performed, and the values of the mean individual biomass (MIB) were calculated. To determine the correlation between mean individual biomass and abundance of various ecological groups of carabid beetles, the Spearman rank correlation coefficient was calculated. The assemblages of ground beetles living in the Norway spruce forests in north-eastern Poland were characterised by quite large species richness (44 species in total). There were significant differences in species richness among the different ages of Norway spruce forests. The oldest Norway spruce stands (70–80 years old) had a smaller number of species and specimens of ground beetles as well as the highest MIB values in comparison with the younger spruce forests A and B. The results revealed that high MIB values were positively correlated with the presence of large ground beetle species with higher moisture requirements. Lower values of the MIB index were due to the presence of smaller open habitat macropterous species, with the spring type of breeding and associated with open areas.

Introduction

In the late 18th century, forests in Poland covered around 40% of the country’s area, and declined to 20.8% in 1945, soon after World War Two (Zajaczkowski et al., 2016). The undesirable environmental changes caused by this process, as well as the great demand for timber were the reasons why the forest cover in Poland increased to 27% between 1945 and 1970. This increase was achieved mainly by growing monoculture forest stands of Scots pine (Pinus sylvestris L.) and Norway spruce (Picea abies (L.) H. Karst) (Krawczyk, 2014). The latter species in its natural range is one of the main forest-forming species. It grows in the mountains of central and southern Europe and also covers extensive areas in the north and east of this continent. Nowadays, spruce in Poland occurs in two areas: the north-eastern (on the lowlands) and the south-western (in the mountains), and makes up around 6% of all forests. The dying of Norway spruce stands observed in recent years may be caused by several factors. The following are mentioned: decreasing soil moisture content, changes in soil chemistry connected with acidification, and shortage of nutrients, e.g., magnesium (Zwoliński, 2003; Małek et al., 2014). Another probable reason why spruce forests are dying is the way they are grown, as a monoculture, where this species forms dense single-storey stands. Monoculture spruce forests are a specific habitat, with poor litter, not very rich understorey and groundcover, and extensive soil surface shading (Holeksa & Szwagrzyk, 2004). Because of the wide concept of ecosystems, studying spruce stands conditions by bioindication is desirable, with the use of appropriate bioindicators.

One of the groups of organisms used for bioindication are ground beetles (Coleoptera: Carabidae). Carabidae make very good bioindicators, and are often used to evaluate changes occurring in agricultural, woodland, and urban habitats (Niemelä, Langor & Spence, 1993; Niemelä, Koivula & Kotze, 2007; Pizzolotto et al., 2018). They are also one of the simple indicators used in descriptions of the development of succession in forests (Szyszko, 1983; Szyszko, 1991; Butterfield, 1997; Skłodowski, 2009). Assemblages of ground beetles dwelling in pine forests are quite well recognised (Schwerk & Szyszko, 2007; Skłodowski, Bajor & Trynkos, 2018). Numerous studies provide the characteristics of ground beetle assemblages in pine forests of different ages, growing on different types of soil and using different forest management practices or having the understorey composed of different deciduous species (Szyszko, 1990; Jukes, Peace & Ferris, 2001; Finch, 2005; Taboada et al., 2008; Barsoum et al., 2014; Lange et al., 2014; Kosewska et al., 2019). Having explored the complex relationships between ground beetle assemblages and the pine forests they inhabit, it became possible to develop a synthetic indicator that describes these dependencies (Szyszko, 1990). The mean individual biomass (MIB) of carabid fauna has been proposed as an indicator of the stage of succession in forests and is now used in research and forest management (Szyszko et al., 2000; Schwerk & Szyszko, 2007; Instrukcja ochrony lasu, 2012; Skłodowski, Bajor & Trynkos, 2018). Regarding spruce forests growing in North America, the Fennoscandian Peninsula, and most of Europe, the research results concerning this group of beetles have been well documented (Reeves, Dunn & Jennings, 1983; Abildsnes & Tommeräs, 2000; Elek, Magura & Tóthmérész, 2001; Jukes, Peace & Ferris, 2001; Koivula & Niemelä, 2002; Lange et al., 2014). Attention was drawn, inter alia, to the influence of Norway spruce monocultures on the reduction of the characteristic ground beetle species of the native forests (Elek, Magura & Tóthmérész, 2001) as well as on the emergence in spruce stands of species with a wide ecological valence (Šustek, 2013). Some authors (Baguette & Gérard, 1993; Magura, Tóthmérész & Bordán, 2000) point out that considerable changes occur in ground beetle assemblages as Norway spruce stands age. In contrast, we lack such reports on ground beetles in Polish spruce forests. Should the species and quantitative composition of Carabidae assemblages in Polish spruce forests be known, such information might be useful for the determination of the advancement of forest succession, and the evaluation of the condition of a given forest habitat, especially in view of the progressing death of Norway spruce stands and risk of increased presence of pests.

The study reported in this article aimed to determine the abundance and species composition of ground beetle assemblages occurring in spruce forests in north-eastern Poland. There are no comprehensive studies on the structure of ground beetle assemblages in Norway spruce forests in Poland, but the state of knowledge of Carabidae in pine forests is very well known. Therefore, in our research we compare whether the Carabidae assemblages of these two types of stands (pine and spruce) are similar, assuming that the assemblages in Norway spruce stands react in a similar way. Based on literature data on pine forests, the hypothesis was that the ground beetle assemblages in the studied spruce forest differ from one another in relation to species richness, abundance, biodiversity, MIB and life traits, depending on the age of the forest. It was assumed that older Norway spruce forests, as demonstrated for pine forests, would have:

—lower ground beetle species richness and abundance,

—higher values of the MIB and biodiversity indices,

—a greater share of large forest zoophagous species accompanied by a decreasing percentage of open-space hemizoophagous species.

Materials and methods

Study area

- The research covered the area of 4 Forest Subdistricts in north-eastern Poland: Sosnowo subdistrict (Skrwilno Forest District) in 2016 (plots: A1, A2–area: 1.0 ha); Zielony Dwór subdistrict (Giżycko Forest District) in 2015 (plots: B1, B2–area: 1.39 ha; A3, A4–area: 1.83 ha); Maruny subdistrict (Wipsowo Forest District) in 2018 (plots: B3, B4:–area: 3.00 ha) and Przejmy subdistrict (Przasnysz Forest District) in 2015 (plots: C1, C2, C3, C4–area: 5.4 ha) (Fig. 1). Letters confirming the consent to make the area available by the National Forest Holding State Forests, sent to the PeerJ publisher. Areas with a dominant share of Norway spruce (Picea abies (L.) H.Karst) (70–100%), representing 3 age categories, were selected:

Figure 1 Location of the study area, with distribution of sampling sites and transect of traps ((A) 20–30 years; 40–50 years; 70–80 years).

A—young: 20–30 years old (plots: A1, A2, A3, A4),

B—middle-aged: 40–50 years old (plots: B1, B2, B3, B4),

C—old: 70–80 years old (plots: C1, C2, C3, C4).

More information concerning the research sites is given in Table 1. Detailed characteristics of the research areas were developed on the basis of data provided by the Forest Data Bank (2023) and on the basis of our own observations. In each age category, 4 plots treated as replications were established. In each research plot, 5 Barber traps were set up, which were plastic containers measuring nine cm in diameter and 13 cm in height. The traps were filled with a preservative liquid and submerged into the soil to be level with the ground surface (Kotze et al., 2011). Traps in each plot were spaced along a transect, at a distance of 10 m from each other. The transects were placed in the central part of the stands; the distances between the transects and the arrangement of traps are shown in Fig. 1.

Table 1 Description and characteristics of spruce forests.

Age variant	District	Plot	Age (years)	Height above sea level (m)	Soil	Type of forest	Share of spruce (%)	Canopy (%)	No of treatments*	
A	Sosnowo	A1	28	135	haplic brunic arenosol	mixed fresh coniferous forest	100	100	1	
A2	28	135	haplic brunic arenosol	mixed fresh coniferous forest	100	100	1	
Zielony Dwór	A3	28	151	haplic luvisol	fresh forest	80	100	1	
A4	28	151	haplic luvisol	fresh forest	80	100	1	
B	Zielony Dwór	B1	43	150	haplic luvisol	fresh forest	80	90	2	
B2	43	150	haplic luvisol	fresh forest	80	90	2	
Maruny	B3	48	139	albic brunic arenosol	mixed fresh forest	90	90	2	
B4	48	139	albic brunic arenosol	mixed fresh forest	90	90	2	
C	Przejmy	C1	78	120	mucky soils	moist forest	70	80	7	
C2	78	120	mucky soils	moist forest	70	80	7	
C3	78	119	mucky soils	moist forest	70	80	7	
C4	78	119	mucky soils	moist forest	70	80	7	
Notes.

* Treatments: cleaning; thinning.

Twenty traps were set up in each age variant of spruce forest, that is, 60 traps in the entire experiment (3 variants × 4 replications × 5 traps). A trap served as an elementary sampling unit. Traps were placed in the ground in spring (May) and emptied every two weeks, simultaneously replenishing the preservative liquid, and the species of captured beetles were identified (Hůrka, 1996). The traps were removed in autumn (October), and the duration of their exposure (days) for each of the research areas was as follows: Przejmy (135), Zielony Dwór (160), Sosnowo (153) and Maruny (141).

Life traits of carabids

Assemblages of ground beetles in the analysed age categories of spruce forests were compared, taking into account the abundance, species richness, and values of the Shannon H’ species diversity index. The species-identified beetles were classified in terms of body size and food preferences, breeding strategy, preferred habitat, habitat moisture content, and ability to fly (Meijer, 1974; Lindroth, 1985; Lindroth, 1986; Hůrka, 1996; Aleksandrowicz, 2004) (Table 2). Regarding the life traits, the captured Carabidae representatives were divided into large zoophages (Lz)—predatory species with a body length of more than 15 mm, small zoophages (Sz)—also predators but with a body length of no more than 15 mm, and hemizoophages (Hz)—ingesting mixed food. Concerning the breeding strategy, the ground beetles were classified as spring (Sb) and autumnal (Ab) types of breeding. Depending on habitat preferences, the following groups of ground beetles were distinguished: forest (F), open area (Oa), eurytopic (Eu), and wetland (Wet) species. In relation to moisture preferences, the distinguished groups of ground beetles were: hygrophilous (H), mesophilous (M), and xerophilous (Xe). The ability to fly was also considered, giving rise to the following division: able to fly, with developed wings, i.e., macropterous species (Ma), species with reduced wings, not flying, i.e., brachypterous (B), and dipterous (D), i.e., those which can be found to present either of the forms of wing development.

Table 2 Abundance, ecological classification and species name abbreviations used in the RDA analysis of ground beetles in age variants of spruce forests.

 	Abbreviations	Ecological characteristic	Individuals / Percentage	
Species			A*	B	C	
			A1, A2	A3, A4	B1, B2	B3, B4	C1, C2	C3, C4	
 	 	 	n	%	n	%	n	%	n	%	n	%	n	%	
Amara bifrons (Gyllenhal,1810)	A_bifr	Oa/Xe/Hz/Ab/Ma	2	0.3											
A. brunnea (Gyllenhal, 1810)	A_bru	F/M/Hz/Ab/Ma	4	0.6							4	0.8	2	0.5	
A. communis (Panzer, 1797)	A_com	Oa/M/Hz/Sb/Ma	1	0.1											
A. similata (Gyllenhal,1810)	A_simi	Oa/M/Hz/Sb/Ma	1	0.1											
Badister lacertosus Sturm, 1815	Ba_lace	F/M/Sz/Sb/Ma					2	0.4			1	0.2			
Calathus ambiguus (Paykull,1790)	Ca_amb	Oa/Xe/Sz/Ab/Ma	1	0.1											
C. erratus (Sahlberg, 1827)	Ca_err	F/Xe/Sz/Ab/D	2	0.3											
C. fuscipes Goeze, 1777	Ca_fusc	Oa/M/Sz/Ab/B	26	3.9					3	0.5					
C. melanocephalus (Linnaeus, 1758)	Ca_mela	Oa/M/Sz/Ab/B	3	0.4							9	1.9	13	3.4	
C. micropterus (Duftschmid, 1812)	Ca_micr	F/M/Sz/Ab/B	181	26.3			2	0.4	14	2.3	19	4.0	17	4.5	
Carabus arvensis Herbst,1784	C_arv	F/Xe/Lz/Sb/B	89	12.9							3	0.6			
C. cancellatus Illiger,1798	C_canc	Eu/M/Lz/Sb/B	3	0.4	18	4.1	12	2.3			3	0.6	3	0.8	
C. convexus Fabricius, 1775	C_conv	F/Xe/Lz/Sb/B	12	1.8											
C. coriaceus Linnaeus, 1758	C_coria	F/M/Lz/Ab/B							1	0.2					
C. glabratus Paykull, 1790	C_glab	F/M/Lz/Ab/B	3	0.4					1	0.2	4	0.8	6	1.6	
C. granulatus Linnaeus, 1758	C_gran	Wet/H/Lz/Sb/B	1	0.1	12	2.8	4	0.8	1	0.2			5	1.3	
C. hortensis Linnaeus, 1758	C_hort	F/M/Lz/Ab/B	118	17.2	215	49.0	236	46.0	108	18.0	229	48.0	172	45.0	
C. marginalis Fabricius, 1794	C_marg	F/M/Lz/Sb/B	2	0.3					5	0.8					
C. nemoralis O.F.Muller, 1764	C_nemo	Eu/M/Lz/Sb/B	1	0.1	19	4.4	35	6.8	23	3.8	28	5.9	19	5.0	
C. violaceus Linnaeus, 1758	C_viol	F/M/Lz/Ab/B	10	1.5					11	1.8	11	2.3	15	3.9	
Clivina fossor (Linnaeus,1758)	Cl_fos	Oa/M/Sz/Sb/D					1	0.2							
Cychrus caraboides (Linnaeus, 1758)	Cy_cara	F/M/Lz/Ab/B			12	2.8	8	1.6			10	2.1	2	0.5	
Harpalus laevipes Zetterstedt, 1828	H_leav	F/M/Hz/Sb/Ma	1	0.1	5	1.1	3	0.6	5	0.8	10	2.1	8	2.1	
H. rufipes (De Geer, 1774)	H_rufipe	Oa/M/Hz/Ab/Ma	4	0.6					3	0.5					
H. tardus (Panzer, 1797)	H_tar	Oa/Xe/Hz/Sb/Ma	3	0.4					2	0.3					
Leistus rufomarginatus (Duftschmid, 1812)	L_rufo	F/M/Sz/Ab/B	2	0.3							2	0.4			
L. terminatus (Hellwig, 1770)	L_term	Wet/H/Sz/Ab/B			1	0.2			1	0.2	1	0.2	1	0.3	
Limodromus assimilis (Paykull, 1790)	Li_ass	F/H/Sz/Sb/Ma			11	2.5	10	2.0			10	2.1	9	2.4	
Loricera pilicornis (Fabricius,1775)	Lo_pili	Wet/H/Sz/Sb/Ma		1	0.2									
Nebria brevicollis (Fabricius, 1792)	N_brev	Eu/M/Sz/Ab/Ma			1	0.2			55	9.0	8	1.7			
Notiophilus biguttatus (Fabricius, 1779)	No_big	F/M/Sz/Sb/D	18	2.7			2	0.4	2	0.3	1	0.2	1	0.3	
N. palustris (Duftschmid, 1812)	No_pal	Oa/M/Sz/Sb/B	1	0.1					8	1.3					
Ophonus rufibarbis (Fabricius,1792)	Op_rufi	Eu/M/Sz/Ab/Ma			1	0.2									
Oxypselaphus obscurus (Herbst, 1784)	Ox_ob	F/H/Sz/Sb/Ma			1	0.2			1	0.2					
Patrobus atrorufus (Strom,1768)	Pa_atr	Wet/H/Sz/Ab/B			4	0.9									
Poecilus lepidus (Leske,1785)	Po_lep	Oa/Xe/Sz/Sb/D	3	0.4											
P. versicolor (Sturm, 1824)	Po_ver	Oa/M/Sz/Sb/Ma	1	0.1			1	0.2	2	0.3	3	0.6	6	1.6	
Pterostichus anthracinus (Illiger,1798)	P_anth	Wet/H/Sz/Sb/D	1	0.1			1	0.2							
P. melanarius (Illiger, 1798)	P_mela	Eu/M/Lz/Ab/D			30	6.9	55	11.0	118	19.0	41	8.6	40	11.0	
P. niger (Schaller, 1783)	P_nige	F/M/Lz/Ab/Ma	86	12.5	66	15.0	87	17.0	175	29.0	22	4.6	29	7.6	
P. oblongopunctatus (Fabricius, 1787)	P_obl	F/M/Sz/Sb/Ma	109	15.8	38	8.7	52	10.0	70	11.0	58	12.0	32	8.4	
P. strenuus (Panzer, 1797)	P_str	F/H/Sz/Sb/D							2	0.3					
P. vernalis (Panzer,1796)	P_ver	Oa/H/Sz/Sb/Ma			1	0.2									
Syntomus truncatellus (Linnaeus,1761)	Sy_tru	Oa/Xe/Sz/Sb/B	1	0.1	 	 	 	 	 	 	 	 	 		
Individuals - Total	 	 	690	100	436	100	511	100	611	100	477	100	380	100	
Notes.

* (A) 20-30 years; (B) 40-50 years; (C) 70 - 80 years.

** Hz- hemizoophages, Sz- small zoophages, Lz- large zoophages, Ab- autumn breeders, Sb- spring breeders, F- forest, Oa- open area, Eu- eurytopic, Wet - wetland, M- mesophilous, Xe- xerophilous, H- hygrophilous, Ma- macropteric, B- brachypteric, D- dipteric.

Data analysis

Moran’s I index was calculated (Rangel, Diniz-Filho & Bini, 2010) to measure spatial autocorrelation of carabid abundance in the sampling plots of the studied areas. The distribution of the abundance of carabid beetles along the spatial arrangement of sampling plots was not autocorrelated (Moran’s I = 0.264, p = 0.117).

The mean individual biomass (MIB) of the Carabidae was calculated using the formula of Szyszko et al. (2000), which describes dependencies between the body length and biomass of beetles. Evaluation of the distribution of the abundance and species richness of Carabidae as well as the abundance of the ecological groups, mentioned earlier, in the analysed variants was supported by the Shapiro–Wilk W test. Data characterised by unimodal distribution were assessed using the generalised linear model (GLM), which included Poisson’s data distribution. Groups of means not statistically different were assigned the same lettered index: a, b (Bonferroni test). Data with normal distribution were examined using the ANOVA test. Spearman’s correlation rank coefficient (rs) was derived to determine the correlation between values of the MIB index and the abundance of specific trophic groups of ground beetles.

To visualize the differences of ground beetle assemblages living in the analysed age variants, non-metric multidimensional scaling (NMDS) was performed, using the Bray-Curtis similarity index. This method enabled evaluation of the ground beetle assemblages based on the similarity between individual beetles and by analysing data concerning the number of captured species and specimens. Samples with a high degree of similarity are plotted close to one another in the NMDS diagram. An assessment of the significance of differences between the analysed assemblages using the NMDS method was performed using the ANOSIM non-parametric statistical test (Clarke, 1993). Ordination methods were used to assess relationships between the presence of Carabidae species and their specific ecological groups versus the distinguished environmental variables (forest age, canopy, humidity, fertility, MIB, and the number of treatments, such as stand cleaning and thinning) (ter Braak & Šmilauer, 1998). The RDA method (Redundancy Analysis) was used because the distribution of the analysed data was linear (SD = 2.09). This type of analysis arranges samples in order along a gradient represented by axes of the ordination diagram, in our case based on data concerning the species composition of ground beetle assemblages. The significance of the environmental variables was tested using the Monte Carlo test. The Jackknife 2 estimator was used for abundance data (using EstimateS v. 9.1.0 statistical software), and the species accumulation curves were calculated to assess the adequacy of the sampling efficiency (Zahl, 1977; Colwell, 2005).

All statistical calculations and their graphical presentation were supported by the following programmes: Statistica 13.3 (Shapiro–Wilk W test, GLM, ANOVA, Spearman’s correlation rank coefficient), Canoco 4.51 (RDA), and Past 2.01 (NMDS and ANOSIM).

Results

During the study, 3 105 specimens belonging to 44 carabid species were captured (Table 2). Species accumulation curves for individual transects in the three years of the study confirmed that the sampling effort was adequate (Fig. 2). The ANOSIM analysis indicated significant differences (R = 0.31, p < 0.01) between the carabid assemblages inhabiting the three age categories of Norway spruce stands; these significant differences are shown on the NMDS diagram (Fig. 3). The dominant species in the analysed forest stands was Carabus hortensis, which composed 34.72% of the total species captured. Other numerous species were: Pterostichus niger (14.98%), Pterostichus oblongopunctatus (11.56%), Pterostichus melanarius (9.15%), and Calathus micropterus (7.50%). The remaining species of ground beetles did not exceed a 5% share in the examined assemblages. The greatest number of species (40) were caught in the young Norway spruce forest classified as A (up to 30 years old), while in the middle-aged spruce forest (B) there were 28 species, and in the old Norway spruce stands (C) there were the smallest number, 22 (Table 2). The GLM analysis confirmed significant differences in the average number of ground beetle species caught between the three age categories of Norway spruce stands. The post-hoc Bonferroni test arranged these average values into three uniform groups (Fig. 4). Analysis of the number of individuals showed that the smallest average catch rate of carabids was found in 80-year-old Norway spruce stands (3.9 ± 0.26 individuals on average). Significantly more individuals were captured in the young and middle-aged stands (A, B), where their mean number was on a comparable level (Fig. 4, Table 3). In contrast the highest MIB value was found in the assemblages inhabiting the old stands (C) (Fig. 4, Table 3). There were no significant differences in the H’diversity index values in the studied Norway spruce stands of different age classes (Table 3).

Figure 2 Expected number of carabid species caught in the studied spruce forests using the Jackknife estimator (±SD) of species richness.

Figure 3 Diagram of non-metric multidimensional scaling (NMDS) for carabid assemblages in relation to the age classes of spruce forests ((A) 20–30 years; (B) 40–50 years; (C) 70–80 years).

Figure 4 Mean numbers of species, individuals, MIB and percentages of Carabidae habitat types statistical significant in the analyzed Norway spruce stands.

An asterisk (*) indicates that charts followed by the same letter do not differ.

Table 3 Results of statistical tests (GLM, ANOVA) and values of Spearman’s rank correlation coefficient for species richness, abundance, diversity and MIB values of Carabidae ecological groups within age classes of spruce forests.

Specifications	Means for the analysed age	Wald’s statistic	ANOVA F value	p	rs Spearman p < 0.05	
	classes of spruce forests					
	A	B	C					
Individuals	5.12	5.10	3.90	45.65	–	0.00	n. s.*	
Species	2.78	2.43	1.50	84.75	–	0.00	n. s.	
Shannon H’	0.78	0.75	0.71	0.07	–	0.96	−0.80	
MIB (mg)	273.82	286.67	341.84	173.1	–	0.00	–	
**Eu	3.65	14.90	7.10	138.18	–	0.00	n. s.	
F	49.25	39.85	33.85	58.29	–	0.00	n. s.	
Oa	2.40	1.00	1.55	11.51	–	0.00	−0.61	
Wet	1.00	0.35	0.35	9.08	–	0.01	0.29	
H	1.65	1.00	1.30	3.17	–	0.21	0.38	
Xe	5.65	0.10	0.15	69.01	–	0.00	−0.58	
M	49.00	55.00	41.40	–	3.34	0.04	n. s.	
Hz	1.05	0.65	1.20	3.26	–	0.20	n. s.	
Lz	34.85	44.00	32.10	41.70	–	0.00	0.34	
Sz	20.40	11.45	9.55	93.60	–	0.00	−0.87	
Ab	38.60	43.90	32.85	31.61	–	0.00	n.s.	
Sb	17.70	12.20	10.00	46.33	–	0.00	−0.55	
B	36.70	23.65	28.60	57.85	–	0.00	n.s.	
D	2.70	9.05	4.15	76.28	–	0.00	n.s.	
Ma	16.90	23.40	10.10	100.78	–	0.00	−0.58	
Age of forest	<80	<50	<30	–	–	–	0.26	
(years)	 	 	 	 	 	 	 	
Notes.

* n.s., non significant.

** Hz, hemizoophages; Sz, small zoophages; Lz, large zoophages; Ab, autumn breeders; Sb, spring breeders; F, forest; Oa, open area; Eu, eurytopic; Wet, wetland; M, mesophilous; Xe, xerophilous; H, hygrophilous; Ma, macropteric; B, brachypteric; D, dipteric.

Regarding habitat preferences in the three age categories of Norway spruce stands, the assemblages of ground beetles were composed mostly of forest species. The highest average number of individuals representing this class of ground beetles was collected in the young spruce forest (A)—49.25 ± 4.25 individuals on average (Table 3). An increase in the age of the forest did not generate a higher mean abundance of forest species of ground beetles (B—39.85 ± 3.70 and C—33.85 ± 3.69), but they still accounted for over 70% of the ground beetle assemblages of these stands (Fig. 4). The second most numerous group consisted of eurytopic species, most abundant in the middle-aged and old forests (Table 3, Fig. 4). Open-area and wetland beetles composed the least numerous groups, presenting a similar distribution in all three age categories of forest (Table 3). Our analysis of ground beetle moisture preferences revealed that mesophilous beetles were the dominant group, and they constituted between 86% (A) and 98% (B) of the studied ground beetle assemblages. The youngest Norway spruce stands (A) comprised the highest percentage of individuals classified as xerophilous ones (Table 3). Concerning the trophic preferences, the examined age categories of Norway spruce stands were populated by a prevalent share of predatory ground beetles (between 65 and 77%). Large zoophagous beetles were most frequently encountered in the middle-aged spruce forest (mean 44.00 ± 4.10). In the other two age categories (young and old), the number of large zoophages was lower (Table 3, Fig. 4). Small zoophages were the most abundant in the young forest (A). Their abundance decreased in older forests (B, C). The dominant type of breeding among the ground beetles was the autumnal strategy, with the most numerous representatives found in the middle-aged (B) and old (C) forests. In the case of spring breeders, the opposite situation was observed; they were significantly most abundant in the young stand (A). As far as dispersion possibilities are concerned the most brachypterous individuals appeared in the young (A) and old (C) Norway spruce forests. The smallest number of beetles with no ability to fly was found in the middle-aged spruce forest (B). In this age category forest, the dominant beetles were winged ones (macropterous and dipterous) (Table 3, Fig. 4).

To determine the correlations between MIB values in the analysed age ranges of spruce forests and the abundance of the identified ecological groups of ground beetles, values of Spearman’s rank correlation coefficient (rs) were calculated. Higher MIB values were significantly positively correlated with the ascending age of spruce forests (rs = 0.26), and with the occurrence of large zoophagous beetles (rs = 0.34) and the group of ground beetles having a high demand for moisture, i.e., hygrophilous (rs = 0.38) and wetland species (rs = 0.29) (Table 3). Low values of the MIB index were correlated with the presence of numerous small zoophagous beetles (rs = −0.87). A significant correlation, in this case, was also found with the numerous occurrence of open area (rs = −0.61), macropterous (rs = −0.58) and xerophilous (rs = −0.58) beetles, and spring breeders (rs = −0.55). Higher values of the Shannon H’ index were also correlated with a decrease in the MIB index values (rs = −0.80) (Table 3).

Further data concerning the ground beetle assemblages are shown on the RDA diagram (Fig. 5). The RDA analysis presents the distribution of ground beetle species and indicates their ecological groups along environmental gradients, represented by two main canonical axes, describing 67.8% of the variance. The redundancy analysis RDA showed that the middle-aged stands (B) , situated in two different forest habitats (also different in terms of humidity) are grouped in one part of the ordination diagram. Similar regularity is observed for most objects located in the young stand (A), which are grouped in the upper, left-hand quarter of the diagram and correlate with the second ordination axis, in addition to which they are also localised in different types of forest habitat. Analysis of this part of the diagram also indicates that high shading of soil (which is canopy dependent) was correlated with the presence of open area species (Oa), small zoophagous species (Sz) with the spring type of breeding (Sb), and xerophilous (Xe) species.

Figure 5 Diagram of the redundance analysis (RDA) illustrating dependences between captured species of Carabidae, ecological groups which they belong to, and distinguished environmental variables.

(Abbreviations are given in Table 2; (A) 20–30 years; (B) 40–50 years; (C) 70–80 years).

Further analysis of the RDA diagram shows that the first ordination axis correlated with the presence of large zoophagous species (Lz), which is also linked to high values of the MIB. This axis is also strongly correlated with habitat humidity and to the presence of hygrophilous species (H) and species associated with wet areas (Wet), e.g., Carabus granulatus and Oxypselaphus obscurus. The fertility of the habitat, based on the various types of forest habitats studied, was not correlated with the first two ordination axes describing most of the ground beetle assemblages.

Discussion

The species richness obtained in this study on ground beetles in spruce forests was relatively high, although it needs to be added that this is the total of ground beetles caught in four localities situated in two macroregions of Poland (the Masurian Lakeland, and the Chełmińsko-Dobrzyńskie Lake District) (Kondracki, 2011). Considering the above, the number of species caught was similar to the reported number of species in spruce forests of Finland (Koivula & Niemelä, 2002; Koivula et al., 2019), spruce- pine forests of Canada (Niemelä, Langor & Spence, 1993), spruce forests of Slovakia (Macko, 2016), and mixed forests of Germany (Lange et al., 2014). There were differences in the average number of ground beetle species in the examined Norway spruce forests, with a tendency towards smaller species richness in older stands (Fig. 4). The same tendency is seen in the number of beetles caught.

The NMDS analysis showed significant differences between ground beetle assemblages from the three age classes of Norway spruce stands (Fig. 3). Observed differences in the assemblages of ground beetles were mainly due to a more numerous presence of large forest species (C. hortensis (46.5% in total), C. violaceus) identified in old stands (C), large forest and eurytopic species in middle-aged spruce stands (B) (P. niger, P. melanarius) as well as small and large forest zoophages in young stands (C) (Calathus micropterus, C. arvensis). The frequent occurrence of C. hortensis, not only in the oldest but also in younger Norway spruce stands is not an unusual finding. This species, in Scandinavia, is characteristic of deciduous and mixed forests, and in central Europe it appears in large numbers in coniferous forests (Lindroth, 1985). At the same time, C. hortensis is considered to be typical of mature tree stands (Niemelä, Koivula & Kotze, 2007). The more numerous appearance of C. violaceus in Norway spruce stands over 80 years old is worth noting in the context of the observed decline in its presence in Polish forests (M. Nietupski, 2020, unpublished data). An interesting finding is the frequent presence of P. melanarius and C. nemoralis in the spruce forests covered by our study. These are eurytopic species, which also appear in forests. However, they belong to the group of deciduous forest specialists, characteristic of native forests (Magura, Tóthmérész & Bordán, 2000). P. melanarius is not a species that lives in abundance in spruce forests (Lynikiene, 2006; Schwerk & Szyszko, 2007), but it is more abundant in younger forests (Tarwacki, 2012).

In the young forest (A), the dominant ground beetles were typical forest zoophagous species with poor dispersal power (brachypterous): C. micropterus and C. arvensis. P. oblongopunctatus also appeared in large numbers (Table 2). The two former species were described by Magura, Tóthmérész & Bordán (2000), in a study on ground beetle assemblages in managed and unmanaged spruce plantations, as species associated with the presence of deciduous trees. These authors also suggested that tree thinning in managed spruce plantations induces the growth of a leafy understory, which leads to the accumulation of leafy litter. In such habitats, the frequent presence of P. oblongopunctatus has been noted.

Analysis of ground beetle assemblages in terms of their ecological characteristics can be useful in assessing the advancement of the forest succession process. Assemblages of carabid beetles associated with early phases of the forest succession process are distinguished by a relatively large share of small species with large dispersion power, often connected with open space areas. In advanced stages of succession, large, forest-type species with lesser dispersion power begin to appear (Magura, Elek & Tóthmérész, 2002; Niemelä, Koivula & Kotze, 2007). The young spruce forests in our research were distinguished by the presence of Carabidae species classified as an open area species being spring breeders, xerophilous, small zoophagous, and hemizoophagous. Although the mentioned groups were not numerously represented, the ordination diagram (RDA) clearly indicates that they are associated with young spruce forests (A). The above ecological description of species composing assemblages of ground beetles is typical of forests at an early stage of forest succession (Buddle et al., 2006; Skłodowski, 2017; Kosewska et al., 2018). It was only in young Norway spruce stands (besides the typical forest species that appeared in all tree stands, i.e., Amara brunnea) that open area species of the genus Amara were detected (Amara bifrons, Amara communis, and Amara similata). These species are connected with open areas (meadows, agrocenoses, etc.), and have uniquely strong dispersal power owing to their ability to fly (Hůrka, 1996). A similar relationship was determined concerning two species of the genus Harpalus: Harpalus rufipes and Harpalus tardus. This group of hemizoophagous species associated with open areas is capable of inhabiting coniferous forests at the early stages of succession, which may be explained by their ability to find plant food in these habitats (Kosewska et al., 2019). The conditions which are suitable for the development of these species disappear as a forest grows older (disappearance of plant food). The number of early-succession species decreases when the tree canopy closes, which is when the forest is 20 to 30 years old (Niemelä, Koivula & Kotze, 2007).

One of the most popular indicators used to identify stages in the succession of forests is the mean individual biomass (MIB) index (Szyszko, 1983; Szyszko et al., 2000; Cárdenas & Hidalgo, 2007; Jelaska, Dumbović & Kučinić, 2011). The results obtained in our study indicate a growth in the MIB values determined for ground beetles in older spruce forests (Fig. 4). The calculated values of Spearman’s rank correlation coefficient (Table 3) indicate that high MIB values are related to the presence of species classified as large zoophages, having high moisture requirements (hygrophilous and wetland species). Low MIB values were significantly correlated with the higher species diversity (Shannon H’) of Carabidae, the frequent presence of small zoophagous, xerophilous species, and open area species with high dispersal capability (macropterous). The presence of species having specific ecological traits in younger and older age classes of spruce forests is closely similar to the situation described by many other authors (Szyszko, 1983; Magura, Elek & Tóthmérész, 2002; Niemelä, Koivula & Kotze, 2007; Skłodowski, 2009). In this study a certain divergence was noted in the decreasing abundance within the species typical of forest habitats with the growing age of the spruce forests. The most probable reason was the frequent occurrence in young forests (A) of C. micropterus and C. arvensis (Table 2). This might be attributed to the fact that all the research sites are habitats typically inhabited by the forest species of ground beetles, and the more frequent presence of certain species in one of the age classes of forests might be due to some specific habitat-related characteristics (e.g., the type of microhabitats) (Pearce & Venier, 2006).

Soil humidity and fertility are also factors affecting Carabidae assemblages, which was also confirmed in many studies (i.e., Skłodowski, 2014; Macko, 2016; Ludwiczak, Nietupski & Kosewska, 2020). Šustek et al. (2017) highlighted that low humidity reduces the activity of ground beetles and their chance to find sufficient prey to survive and complete their development.

Conclusions

Assemblages of ground beetles in spruce forests in north-eastern Poland are represented mainly by large forest zoophages, regardless of the age of the stands. As in other forest types (i.a. pine forests), old spruce stands are characterized by lower abundance and species richness but higher mean individual biomass (MIB) relative to younger stands. Spruce stands of different ages are characterized by different species composition of Carabidae; however, this is also related to other factors, such as forest type, fertility, and soil humidity. Thus, further detailed studies are required to know the precise shaping of Norway spruce forest ground beetle assemblages.

Supplemental Information

Supplemental Information 1 Raw data

Click here for additional data file.

Supplemental Information 2 Statistical descriptions

Click here for additional data file.

We would like to thank Magdalena Obidzińska, Joanna Skurapowicz, Maciej Gągała and Jacek Maczyszyn for help with material collection. We are grateful to Dr Robert Lee for improving the English.

Additional Information and Declarations

Competing Interests

Author Contributions

Field Study Permissions

Data Availability

The authors declare there are no competing interests.

Mariusz Nietupski conceived and designed the experiments, performed the experiments, analyzed the data, prepared figures and/or tables, authored or reviewed drafts of the article, and approved the final draft.

Agnieszka Kosewska performed the experiments, analyzed the data, authored or reviewed drafts of the article, and approved the final draft.

Emilia Ludwiczak performed the experiments, analyzed the data, authored or reviewed drafts of the article, and approved the final draft.

The following information was supplied relating to field study approvals (i.e., approving body and any reference numbers):

The National Forest Holding State Forests approved the study.

The following information was supplied regarding data availability:

The raw data are available in the Supplemental Files.

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
