# Peer review of "Ground beetle assemblages inhabiting various age classes of Norway spruce stands in north-eastern Poland"

_PeerJ, doi:10.7717/peerj.16502_

## Round 0.1 · original submission · Major Revisions

Dear Dr. Nietupski and colleagues:

Thanks for submitting your manuscript to PeerJ. I have now received three independent reviews of your work, and as you will see, the reviewers raised some concerns about the research. Despite this, these reviewers are generally optimistic about your work and the potential impact it will have on research studying ground beetle biology and systematics. Thus, I encourage you to revise your manuscript, accordingly, taking into account all of the concerns raised by both reviewers.

All three reviewers provided many comments for concern that, when addressed, will greatly improve your manuscript. Please check over the English and writing.

Please ensure that your figures and tables are clear and stand-alone. Please elaborate on the impact of your findings in the Discussion and try to add the most relevant literature per the recommendations of the reviewers. Please eliminate redundancy across the manuscript, especially between the Discussion and conclusion Sections. Also, please consider the concerns of reviewer 3 regarding the conclusions that you draw from your data.

Suggestions for removal or addition of data should especially be addressed in your revision.

I look forward to seeing your revision, and thanks again for submitting your work to PeerJ.

Good luck with your revision,

-joe

Reviewer 1 ·

Basic reporting

The manuscript is written in good English. The introduction provides with a sufficient overview of the scientific background of the paper with well-chosen references. However, I propose a slight modification of the title: The authors state in lines 74-75 that” in contrast, we lack such reports on ground beetles in Polish spruce forests”. Therefore, I propose “Ground beetle assemblages of Polish spruce forests depending on the age class of the stand” as title.

Basically, the structure of the manuscript is according to the instructions given by PeerJ. However, the authors mention funding in the Acknowledgements, but according to the PeerJ author-instructions authors should “not acknowledge funders here, there is a separate Funding Statement for that”.

All tables and figures are relevant for the paper. The figures were provided as PDF files. Therefore, it is difficult to assess their original resolution, but they seem to be of good graphical quality. However, figures and their captions should be self-explanatory. Therefore, the authors should add to the captions of figs. 1 and 3 information about what means “A”, “B” and “C”.

In accordance with the PeerJ policy the raw data are provided with the manuscript.

Experimental design

The manuscript is based on original primary research on an biological-ecological research topic, which falls into the general scope of the journal. The basic aim of the study and the hypotheses are well defined (lines 80-88), but please consider my comments at “validity of the findings”. The authors state that there is a lack of reports on ground beetles in Polish spruce forests. Indeed, the majority of studies on ground beetles in Polish coniferous forests is carried out in pine forests. Therefore, the research is meaningful.

The field research was carried out accurately according to standards in the research on ground beetles (Carabidae). Each age class of forest stand was studied in 4 plots treated as replications, thus enabling reasonable statistical analysis. The statistical methods are well chosen in order to study the hypotheses formulated by the authors.

In general the methods are described in sufficient detail. However, in line 103 the authors state “More information about the research sites is given in table 1”. It might be interesting for the reader, from which sources this information was taken, but neither the text nor the caption of the table provide with such information. Regarding the Redundancy Analysis (RDA) information about the values for some environmental variables is missing. With respect to MIB, number of treatments, canopy and forest age this information can be obtained from table 1, but regarding humidity and fertility this is not clear.

Validity of the findings

The manuscript deals with an interesting and important topic. The underlaying data seem to be robust and statistically sound. Therefore, the study provides with valuable knowledge regarding the formation of ground beetle assemblages in Polish spruce forests.

The results are described accurately and are discussed in a though-out manner taking into account an adequate number of references. However, the data were elaborated in four different sub-districts in north-eastern Poland, which differ with respect to soil type and moisture conditions. The authors write in lines 82-83 that the hypothesis was that the ground beetle assemblages “differ from one another depending on the age of the forest, type of woodland habitat, and its humidity”. The focus of results and discussion, however, is strongly on differences due to age (classes). I would expect at least a deeper discussion of the impact of the other factors. For example, the RDA already indicates that the moisture differences seem to have an impact, too. The authors write themselves in the results that the first ordination axis “is also strongly correlated with habitat humidity”. I recommend testing significance of the environmental variables using Monte Carlo permutation tests in order to check, whether forest age is indeed the dominating environmental factor. This would increase the scientific value of the paper.

The conclusions formulated by the author are well stated and related to the initially formulated research questions.

Additional comments

1) Introduction, line 48: Should be “Małek et al., 2014”.

2) Materials and methods, line2 148-149: Should be “…using the formula by Szyszko et al. …”.

3) Results lines 199-205: In line 209 the authors formulate “…the dominant share of ground beetles…”, but from the next lines I understand that they refer to total numbers. The term “share”, however, implies a kind of relation to additional of data, for example a percentage share of a sample. Therefore, this term seems to be misleading.

4) Discussion, lines 259-262: The authors write “the number of species caught was similar to the reported number of species in Finland (Koivula & Niemelä, 2002; Koivula et al., 2019), Canada (Niemelä, Langor & Spence, 1993), Slovakia (Macko, 2016), and Germany (Lange et al., 2014).”. In which habitats in these countries? Spruce forests? Please specify.

5) Discussion, lines 273-274: The authors state that regarding the species Carabus violaceus a decline in its presence in Polish forests is observed. Is this an own observation or is this statement based on literature? If the latter, please cite the respective sources. There are different forest types in Poland and I do not have the impression that this species declines in all forest types (for example pine forests).

6) Discussion, lines 288-291: The sentence seems to be a bit strange formulated, maybe better: “…Carabidae species classified as an open area species being spring breeders, xerophilous, small zoophagous, and hemizoophagous.”.

7) Figure 2: The authors state that they had 60 traps (elementary sampling unit), but in the figure the number of samples is only 55. I assume this is due to trap loss. Please specify.

·

Basic reporting

The manuscript (hereafter ms) topic is relevant and suits well with the standards and the scopes of PeerJ. The ms is well-written the language is clear and concise, the quality of the figures and tables are well-established in terms of typography. However there are some clumsy sentences/parts in the ms, which may request some improvements. The introduction and the applied references are relevant, I missed only the detailed research questions or hypotheses, since this part of the intro is quite descriptive and seems a little bit mechanistic description of the planned work. The submitted raw database suits well with the ms, and eligible for reproducible research initiatives.

Experimental design

The suggested study is a neat contribution to field of forest ecology even though this topic was more relevant and studied some decades ago. I have found no clear research questions or hypotheses in the ms, however the logical concept of the study is feasible and well-documented. In general, the presented study is well -established and rigorous and author are highly controlled the most influential confounding factors during the experiment. The description of the methods and the applied analyses are well-documented I have no doubts that the whole study is reproducible based on the description provided in the ms.

Validity of the findings

The presented results seems concise and logically feasible, and in line with the applied methods and analyses. However the description is quite descriptive and mechanistic, I would expect the biological description of the patterns instead of the statistical one. The discussion is relatively long and inaccurate in some case (see general comments for details). Here I really felt the beneficial role of the clear research question or hypotheses, which might help to organize better the discussion. The conclusion part is completely out of scope, should be removed since a pure repetition of some parts of the discussion, but no implication or perspectives given. Another issue I may suggest to improve is the conceptualization of the presented analyses: 1) In general all the analyses should be unified in a way that the age of the spruce plantation as a factor with three levels should be the main explanatory variable in all analyses including rarefaction and GLM; 2) there are some type of the applied analyses which simply overlap, thus they can be reduced to use only one technique such as NMDS and RDA should be merged or the MIB calculation might be integrated into a updated GLM model as one of the potential response variables.

Additional comments

Line 1 - This is the first case but it should be applied for the whole ms, that the spruce should be updated into the more precise term “Norway spruce”. In addition the scientific name should be appear at least in the intro or site description in the methods.

Lines 14-15 - The coding of the various age classes of the spruce plantation should be updated into young (20-30 year-old plantations), middle-aged (40-50 year-old plantations) and old (70-80 year-old plantations) spruce plantations. This concept should be followed on only in the text but should apply on the figures and the tables as well. Using this age label may improve the readability of the overall ms.
Line 53 - the first sentence is quite clumsy and have no clear message, please rephrase.
Line 54 - should write: “ground beetles (Coleoptera: Carabidae)”

Line 71 - The international comparisons for spruce plantations is not well implemented, there is no clear pattern defined for international (ie. European trend) which is the homogenization in species composition in the ~ old plantations. This trend should be compared with the Polish data.

Line 85-88 - These predictions are quite arbitrary, please suggest prediction that can be falsifiable as suggest by Mood for hypothesis testing. Now the present layout is a direct conversion of the results into predictions which is very redundant and not so elegant.

Line 99 - I missed the information of the rotation of forestry management. This is very information since it will influence the age structure of the spruce plantations, especially for old ones.
Lines 116-119 - the statistical analyses should move into the part under “data analysis” including spatial autocorrelation issues.

Lines 140-145 - This section should move into discussion since it is a reasoning part, in addition please check the proper spelling of the cited authors names for typos in the overall text.

Lines 148-179 - I suggest some conceptual update for the statistical analyses: 1) Multivariate analyses including RDA; 2) Rarefaction curves, 3 GLM model including MIB index as one of the tested response variables. Please present the result in the same order. As I already suggest above all the analyses should be unified in a way that the age of the spruce plantation as a factor with three levels should be the main explanatory variable in all analyses including rarefaction and GLM; 2) there are some type of the applied analyses which simply overlap, thus they can be reduced to use only one technique such as NMDS and RDA should be merged or the MIB calculation might be integrated into a updated GLM model as one of the potential response variables.

Lines 256-264 - This section about the assemblage characteristics suits more in to the results than in the discussion. Here the major message is the homogenization in species composition.

Line 265 - NMDS not show too much differences, the pattern may underline the nestedness of the different-aged plantations.

Table 2 - I may suggest to move this table into ESM, since it is rather descriptive.

Table 3 - Please explain all the abbreviations for all the response variables in the legend.

Figure 1 - In the preview, it seems that the resolution of the map is quite low.

Figure 2 - The panel of the figure is broken between pages

Figure 3 - Please remove this figure, I may suggest an updated multipanel figure including the significant responses from the updated GLM models.

Reviewer 3 ·

Basic reporting

The study examined the effect of forest age on ground beetle assemblage, functional group and mean individual mass in spruce forests by surveying the spruce forest that aged 20-30 years, 40-50 years, and 80 years. Pitfall traps were used and set up at 4 plots of each forest age, which were treated as 4 replicates. The authors showed that higher species diversity was recorded in young forests relative to old growth forests. Most young forest species were characterized by low MIB owing to the presence of macropterous species; while opposite is true in old growth forests. In general, the study is interesting and is important for any effort in biodiversity conservation in temperate regions. However, the work contains a number of major flaw that required rectification before it is considered for acceptance. In addition, the writing is, though, readable, contains numerous awkward sentences and it is not well organized (Pls see my suggestion below). I managed to pick up some and there are still plenty. I suggest the writing should be vetted by fluent English speakers.

Experimental design

Analysis of GLM is not suitable in the work because spatial and ecological traits (different forest sites apparently have different altitude, soil types, forest types and so on) variation in biodiversity may affect the result.

In addition, the study sites included 4 Forest Subdistricts in north-eastern Poland, which
Sosnowo (Forest District Skrwilno) in 2016 (plots: A1, A2 . area: 1.0 ha); subdistrict Zielony Dwór (Forest District in 2015 (plots: B1, B2 . area: 1.39 ha; A3, A4 . area: 1.83 ha);
subdistrict Maruny (Forest District Wipsowo) in 2018 (plots: B3, B4: . area: 3.00 ha). However, the beetle community in the different forests, although same age category) are highly distinct. Thus the data should not be pooled according to forest age. This may cause what being discussed in the discussion are not matched with what results showed. I would suggest the authors to treat and analyze A1 and A2, A3 and A4, B1 and B2, B3 and B4 independently. Obviously, location, surrounding environment, forest types, and so on that may influence the assemblage in a given spruce forest and may be included in the discussion.

Validity of the findings

I am not convinced with the discussion and the conclusion drawn as the fact is not consistent with Table 2 and RDA (see comment below). Pls revise.

Additional comments

L15. Change ‘were submitted for research’ to were investigated
L18. The biodiversity indices
L20. To determine the correlation between mean individual biomass and abundance of carabid beetles, ….
L22. The sentence ‘the assemblages of ground…….’ is vague. Pls state the values of abundance, species richness, and the Shannon H. Index of each site surveyed
L23. There were significant differences in species richness among the different ages of spruce forests
L25. The oldest spruce…. Pls state the values…
L27. The results revealed that high MIB….
L30. the presence of smaller open-habitat macropterous species
L39. the forest cover in Poland increased to…..
L80. The authors made a series of assumption on the assemblages of beetle in old forests. However, beside history of polish forest and beetles used as bioindicators, the introductions did not attempt to lay a research background/foundations that why we should expect to see the differences in beetle community in forests of different ages, beetles’ sizes, as well as reduced species richness.

L108-109 and Fig. 1 pls include scale into the figure and the words are too small to read.
L140-145. this paragraph may be more suitable in Introduction.
L184. Species accumulation curves for individual transects in the 3 years of the study confirmed that the sampling effort was adequate (Figure 2). However, how do we know that sampling effort is sufficient based on the figure 2. In addition, the observed species numbers must be shown in the figures. It is not possible to know how many species number different from estimated sp no. from the Jackknife estimator otherwise.
L153 and L191. I afraid that GLM may not be suitable in determining the differences of beetles’ abundance, species number as well as MIB between forests of different ages. One main reason is that different forest sites apparently have different altitude, soil types, forest types and so on (Table 1). These traits may, to certain extent, affect the beetle community in locale. Thus, I would suggest the authors to consider GLMM, rather GLM, to account the traits as random effect and age as fixed effects. Authors may want to consider to use Poisson distribution for abundance and sp. no. and Gaussian for MIB.
L180-252. It is difficult to follow the result. I suggest authors to separate the results as well as discussion into subsections, for instance, 1. Species diversity and community structure (include anosim). 2. Functional group (include RDA result) 3. Mean individual mass.
L195. ‘3.9 individuals on average’ and throughout the manuscript that each average value should be followed by standard deviation.
Fig. 3. I suggest to change the line chart to bar chart. This is because variable in axis x is not a continuous data.
L231. Higher values of the Shannon H. index were also correlated with a decrease in the MIB index values (rs=-0.80). what is the purpose to carry out the analysis?
L233 – 246. The RDA diagram is not easy to understand. This is because it is not known that what open squares, open circles, and close circles in fig. 5 indicated?

L266. I disagree with the sentence ‘qualitative and quantitative changes in the assemblages of ground beetles identified in the older stands (B and C)…’ This is because authors did not observe the change of assemblage of beetle over time but what authors observed was the differences between forests.
L267-269. My argument is that would the presence of large species in old forest is because of a large species favor deciduous and mixed forest type rather than affected by forest age.

L267-274. The fact is opposite with the finding (Table 2). Based on Table 2, C. hortensis is abundant in young forests (A3 and 4: 215 (49%)). In addition, the number of C. violaceus is low in old growth forest, and C. violaceus does present in young forest (A1 and A2). Similar problems also apply to dispersal ability of beetle, which A1 and A2 contained abundant macroapterous beetles but not in A3 and A4. In further, brachyapterous species is not exclusively present in old growth forests.
Obviously, forest type or surrounding forest may play role in shaping the beetle assemblage.

---

## Round 0.2 · Minor Revisions

Dear Dr. Nietupski and colleagues:

Thanks for revising your manuscript. The reviewers are mostly satisfied with your revision (as am I). Great! However, there are a few remaining concerns to address (per reviewer 1).

Please address these ASAP so we may move towards acceptance of your work.

Best,

-joe

Reviewer 1 ·

Basic reporting

The comments below concern a revised version of the manuscript.

Regarding my comments in the review of the first version I can state that the corrections proposed by me were carried out. The authors modified the title of the manuscript and I very much agree with the new title.

Experimental design

The comments below concern a revised version of the manuscript.

With respect to my comments on the first version of the manuscript the proposed improvements were carried out and unclear aspects were specified.

However, probably due to comments by other reviewers, the formulation of the aim of the study was modified. In my opinion not all of these changes are an improvement: The authors write:

"The study reported in this article aimed to determine the abundance and species composition of ground beetle assemblages occurring in spruce forests in north-eastern Poland. There are no comprehensive studies on the structure of ground beetle assemblages in Norway spruce forests in Poland, but the state of knowledge of Carabidae in pine forests is very well known. Therefore, in our research we compare whether the Carabidae assemblages of these two types of stands (pine and spruce) are similar. Based on literature data on pine forests, the hypothesis was that these ground beetle assemblages differ from one another in relation to species richness, abundance, biodiversity, MIB (Mean Individual Biomass) and life traits, depending on the age of the forest. It was assumed that older Norway spruce forests as well as pine forests would have:
- lower ground beetle species richness and abundance,
- higher values of the MIB and biodiversity indices,
- a greater share of large forest zoophagous species accompanied by a decreasing percentage of open-space hemizoophagous species."

The paragraph cited above suggests to some degree that also data on ground beetles in pine forests are analysed in the paper, but the authors analyse only data elaborated in spruce forests. However, to my understanding the authors use the existing knowledge on ground beetle assemblages in pine stands in order to formulate hypotheses for their studies in spruce stands. If I am right, I suggest to modify the above cited paragraph, maybe as follows:

"The study reported in this article aimed to determine the abundance and species composition of ground beetle assemblages occurring in spruce forests in north-eastern Poland. There are no comprehensive studies on the structure of ground beetle assemblages in Norway spruce forests in Poland, but the state of knowledge of Carabidae in pine forests is very well known. Therefore, in our research we compare whether the Carabidae assemblages of these two types of stands (pine and spruce) are similar, assuming that the assemblages in Norway spruce stands react in a similar way. Based on literature data on pine forests, the hypothesis was that the ground beetle assemblages in the studied spruce forest differ from one another in relation to species richness, abundance, biodiversity, MIB (Mean Individual Biomass) and life traits, depending on the age of the forest. It was assumed that older Norway spruce forests, as demonstrated for pine forests, would have:
- lower ground beetle species richness and abundance,
- higher values of the MIB and biodiversity indices,
- a greater share of large forest zoophagous species accompanied by a decreasing percentage of open-space hemizoophagous species."

Validity of the findings

The comments below concern a revised version of the manuscript.

The authors responded on my comments by adding an additional paragraph to the manuscript. However, they mention that the environmental variable age was tested using the Monte Carlo test, but I cannot find this information in the manuscript. The authors might consider to add this information.

Additional comments

The comments below concern a revised version of the manuscript.

In the revised version of the text improvements proposed by me were carried out and unclear aspect were clarified by the authors in a satisfying manner.

·

Basic reporting

The manuscript (hereafter ms) topic is relevant and suits well with the standards and the scopes of PeerJ. The ms is well-written the language is clear and concise after the revision. The submitted raw database suits well with the ms, and eligible for reproducible research initiatives.

Experimental design

The suggested study is a neat contribution to field of forest ecology even though this topic was more relevant and studied some decades ago. The logical concept of the study is feasible and well-documented. In general, the presented study is well -established and rigorous and author are highly controlled the most influential confounding factors during the experiment. The description of the methods and the applied analyses are well-documented I have no doubts that the whole study is reproducible based on the description provided in the ms.

Validity of the findings

The presented results seems concise and logically feasible, and in line with the applied methods and analyses. The revision improved the discussion and issues on the analyses.

Additional comments

Authors have made a neat revision on the manuscript with detailed replies for each referees. I have no further suggestion for improvement on this submission.

---

## Round 0.3 · accepted · Accept

Dear Dr. Nietupski and colleagues:

Thanks for revising your manuscript based on the concerns raised by the reviewer. I now believe that your manuscript is suitable for publication. Congratulations! I look forward to seeing this work in print, and I anticipate it being an important resource for groups studying ground beetle biology and systematics. Thanks again for choosing PeerJ to publish such important work.

Best,

-joe